# Oncolytic Virotherapy and Microenvironment in Multiple Myeloma

**DOI:** 10.3390/ijms22052259

**Published:** 2021-02-24

**Authors:** Valentina Marchica, Federica Costa, Gaetano Donofrio, Nicola Giuliani

**Affiliations:** 1Department of Medicine and Surgery, University of Parma, 43126 Parma, Italy; valentina.marchica@unipr.it (V.M.); federica.costa@unipr.it (F.C.); 2Department of Medical-Veterinary Science, University of Parma, 43126 Parma, Italy; gaetano.donofrio@unipr.it; 3Hematology, Azienda Ospedaliero-Universitaria di Parma, 43126 Parma, Italy

**Keywords:** oncolytic virotherapy, multiple myeloma, microenvironment

## Abstract

Multiple myeloma (MM) is a hematologic malignancy characterized by the accumulation of bone marrow (BM) clonal plasma cells, which are strictly dependent on the microenvironment. Despite the improvement of MM survival with the use of new drugs, MM patients still relapse and become always refractory to the treatment. The development of new therapeutic strategies targeting both tumor and microenvironment cells are necessary. Oncolytic virotherapy represent a promising approach in cancer treatment due to tumor-specific oncolysis and activation of the immune system. Different types of human viruses were checked in preclinical MM models, and the use of several viruses are currently investigated in clinical trials in MM patients. More recently, the use of alternative non-human viruses has been also highlighted in preclinical studies. This strategy could avoid the antiviral immune response of the patients against human viruses due to vaccination or natural infections, which could invalid the efficiency of virotherapy approach. In this review, we explored the effects of the main oncolytic viruses, which act through both direct and indirect mechanisms targeting myeloma and microenvironment cells inducing an anti-MM response. The efficacy of the oncolytic virus-therapy in combination with other anti-MM drugs targeting the microenvironment has been also discussed.

## 1. Introduction

Multiple myeloma (MM) is an incurable disease characterized by the accumulation of malignant plasma cells in the bone marrow (BM) microenvironment [1]. In addition to genetic and microenvironmental alterations, MM is supported by a high degree of immune dysregulations [2,3]. Indeed, the tight relationship between MM cells and BM microenvironment cells and hypoxia, creates a permissive niche, with impaired dendritic cell (DC) differentiation and maturation, high levels of myeloid derived suppressor cells (MDSCs). Moreover, MM patients display a high percentage of regulatory T cells (Tregs), an unbalanced T helper (Th)1/Th2 cells ratio and altered functionality of natural killer (NK) cells [2]. Furthermore, an increased expression of senescent and exhaustion markers, as programmed death (PD)-1/programmed death-ligand (PD-L)1, cytotoxic T-lymphocyte antigen (CTLA-4) and T cell immunoreceptor with Ig and ITIM domains (TIGIT) has been detected in BM immune-microenvironment of MM patients, leading to the development of monoclonal antibodies (mAbs) targeting these molecules.

Several strategies are currently used in the treatment of MM. The standard of care [4,5] includes a high dose of melphalan followed by autologous stem cell transplantation in eligible patients. Induction therapy before transplantation consists in a combined treatment with bortezomib-thalidomide or lenalidomide-dexamethasone for 4–6 cycles. Recently, the role of maintenance therapy with lenalidomide has been proved to extend the survival of MM patients. In elderly and frail patients, the combination of the proteasome inhibitor (PI) bortezomib with melphalan and prednisone or with lenalidomide and dexamethasone or the combination of lenalidomide and dexamethasone as continuous treatment are considered standard care.

The treatment of relapsed disease includes different combinations with PIs (bortezomib, carfilzomib and ixazomib), immunomodulatory drugs (IMiDs) (lenalidomide and pomalidomide) and dexamethasone. Moreover, both PIs and IMiDs can be combined with mAbs anti-CD38, such as daratumumab. Due to the introduction of these new drugs and transplant techniques, in the past decades, mortality of MM patients has fallen, and 5-year survival has more than doubled particularly in patients aged under 75 years [6]. However, the treatment of patients with refractory relapsed MM remains an urgent medical need due to the high incidence of refractory disease after three lines of treatment. Recently, new therapeutic strategies targeting the microenvironment have been developed, opening new perspectives in the cure of the disease including CAR-T cells, drug-conjugate mAbs and bispecific Abs [7].

Oncolytic virotherapy is an alternative therapeutic technology in cancer treatment that uses natural or genetically engineered viruses as pharmaceutical antitumor agents [8,9,10]. Tumor cells by expressing receptors or adhesion molecules on their cell surface can be targeted by viruses. Indeed, oncolytic viruses are able to selectively infect, transduce and consequently kill cancer cells without affecting normal tissue [11]. The mechanisms of action of oncolytic viruses can be direct, through selective replication within neoplastic cells, or indirect through alterations of the tumor microenvironment that lead to the induction of the antitumor immune response [12,13]. Several oncolytic viruses have shown promising results for the treatment of solid and hematological tumors [14,15,16] and the possibility of using these viruses in tumor immunotherapy has emerged in recent years [17,18]. Researchers are harnessing all the characteristics of viruses, enhancing their effect on cancer cells and awakening and activating the immune system [19].

However, despite encouraging results due to the introduction of several new drugs, recent therapies have not been shown to entirely eradicate MM and overcome drug-resistance [3,20]. Thus, novel treatment strategies are needed. In this context, oncolytic virotherapy is currently used as an immunotherapeutic approach to increase tumor immunogenicity, even in MM [21].

## 2. Direct Mechanisms of Action of Oncolytic Viruses in MM

Tumor cells overexpress adhesion molecules, which bind and mediate the infection of oncolytic viruses [22,23]. Furthermore, the permissive nature of the tumor cell allows the uncontrolled replication of the genetic material and virus propagation within the malignant cell, leading to enhanced cell death [24,25]. The oncolytic viruses studied for MM treatment exploit the same mechanisms. In particular, overexpression of surface molecules, such as CD46 exploited as entry site by several viruses, and mutations in the signaling pathways make MM cells more sensitive to viral infections [26,27]. Figure 1 summarizes the main direct mechanisms.

### 2.1. Human Viruses

The use of human viruses as a therapeutic weapon against cancer dates back to 1897 [19]. The first case reports highlighted the regression of tumors during naturally acquired viral infections [10]. In recent years, they have been genetically engineered to further attenuate their pathogenicity, increase their oncolytic potency and improve their specificity for tumor tissues. In the following paragraphs we describe the main human viruses studied for the treatment of MM.

#### 2.1.1. Measles Virus

It has been reported that MM, as other cancer cells, overexpress CD46 compared to normal cells [28]. CD46 is the major cellular receptor for measles virus (MV) and is responsible for virus attachment, entry and cell killing through cell-cell fusion [29,30,31]. MV is an enveloped negative-sense single-stranded RNA virus in the family *Paramyxoviridae* with known oncolytic property [32,33]. Several studies showed that oncolytic MV replicates selectively in MM cells and its cytopathic effects correlates with CD46 levels on myeloma cells [29,34]. Peng et al. observed an antitumorigenic and antineoplastic activity of MV against human MM cells in xenografts models. Moreover, they demonstrated that MV administrated intravenously causes regression of myeloma disease [34]. Subsequently, Russell et al. reported clinical responses of 2 patients with refractory MM treated with oncolytic MV engineered to express the human thyroidal sodium iodide symporter (NIS). In particular, they demonstrated a decrease in serum free light chain (FLC) levels, a reduction of the percentage of malignant plasma cells and of extramedullary masses in both patients after a single intravenous administration of the virus [35]. A phase I study was conducted to evaluate the MV-NIS safety and maximum tolerated doses of patients with relapsed refractory MM. The selectivity of MV for the tumor has been confirmed and no toxicity has been found. Moreover, one patient achieved complete remission, while the remaining patients reported a decrease of myeloma IgG and a decrease of FLC levels [16]. 

#### 2.1.2. Reovirus 

Another virus with direct activity against MM is the reovirus (RV), a double-stranded RNA virus belonging to the *Reoviridae* family, able to exert potent antitumor activity [36,37]. The attachment and internalization of RV are mediated by junctional adhesion molecule-A (JAM-A) [38]. JAM-A is a transmembrane protein that represents an important component of tight junctions between epithelial and endothelial cells, and plays a role in tight junctions formation with a cytoskeleton [39,40]. Moreover, it has been reported that JAM-A binds directly RV and allows it to entry into cells [41]. The RV-JAM interactions are required for activation of nuclear factor kappa-light-chain-enhancer of activated B cells (NF-κB) and induction of apoptosis during RV infection [38]. Literature data reported that JAM-A increases across the progression of MM disease [42]. Kelly et al. showed that JAM-A expression correlates with increased sensitivity to RV infection in MM cells [43]. Indeed, JAM-A expression promotes RV replication, then inducing MM cell apoptosis. In addition, RV upregulates autophagy and can trigger endoplasmic reticulum (ER) stress during MM oncolysis thereby reducing tumor burden in mouse models [44,45].

#### 2.1.3. Coxsackie Virus

Coxsackie virus (CV) is also used for oncolytic therapy. CV is non-enveloped positive-sense single-stranded RNA virus of which at least 29 strains of CV have been identified [46]. Among these, the most studied and used for virotherapy in solid and hematologic tumors is CV-A21. Its primary receptor, for attachment and internalization in the cell, is intercellular adhesion molecule (ICAM)-1, which in turn interacts with decay-accelerating factor (DAF). ICAM-1 is known to mediate the interaction between MM and stromal cells, thus facilitating MM growth and survival [47]. Moreover, its expression correlates with advanced disease and resistance to chemotherapy [48]. MM cells are known to express ICAM-1 and DAF at high levels, making them the perfect target for CV-A21 oncolytic virus therapy [49]. Indeed, Au et al. demonstrated that CV-A21 induces cytolysis of MM cell lines, and its action is selective for the CD138+ cells from MM patients. Of note, it has reduced cytotoxicity in non-malignant cells. Interestingly, the authors reported the ability of CV-A21 to also eliminate premalignant plasma cells from patients with monoclonal gammopathy of undetermined significance (MGUS), the asymptomatic precursor state of MM, suggesting the use of CV as a preventive treatment [49].

#### 2.1.4. Adenovirus

Adenovirus (AdV) is non-enveloped double-stranded DNA extensively used for gene therapy [50]. Among its different serotypes, AdV5 is the most studied in oncolytic therapy for MM [51,52]. The receptor that allows AdV to enter and infect myeloma cells is still unknown. However, MM cells highly express the coxsackievirus and adenovirus receptor (CAR), which allow the entry of AdVs into mammalian cells and could probably make MM cells more susceptible [53]. On the other hand, the ability of AdV to induce cell death is well documented. Senac et al. reported that AdV5 has a high efficiency of infection in CD138+ cells from MM patients, with a high degree of specificity and selectivity for myeloma cells, thus maintaining the BM microenvironment normal cells unaltered [51]. Another study shows that AdV can be exploited for the CD40 ligand (L)-targeted delivery to MM cells. CD40L is a MM growth inhibitor [54], and binding its CD40 receptor stimulates and activates the immune system [55]. Since CD40 is expressed in different cell types, the use of CD40L vectors must be targeted to avoid non-specific immune activation. In this study, Fernandes et al. demonstrated that AdEHCD40L, an AdV armed with CD40L, inhibits MM cell growth in vitro, inducing apoptosis and cell cycle arrest. Additionally, intratumoral treatment with AdEHCD40L reduced MM cell growth in vivo in a xenograft model, leading to a reduction of tumor volume of up to 54% compared to controls [56]. A gene expression analysis was performed in order to investigate the molecular mechanism of AdEHCD40L-mediated growth inhibition in MM cells. The authors showed that interferon (IFN)-α, interleukin (IL)-8 and chemokine (C-C motif) ligand (CCL)5 were upregulated and Fas and TNF-related apoptosis-inducing ligand (TRAIL) [56]. Study from Wenthe et al. investigated the effect of lokon oncolytic adenovirus (LOAd), a serotype 5/35 chimera able to infect CD46+ cells, including MM cells [57]. Specifically, the authors demonstrated that LOAd infected cells are killed by oncolysis as shown by the detection of viral replication in tested human myeloma cell lines (HMCLs). Moreover, it was reported a decrease of C-X-C motif chemokine ligand (CXCL)10, IL-8 and CCL2 levels and the upregulation of CCL3 post LOAd infection, which could facilitate the recognition of infected MM cells by DCs. In addition, a decreased expression of costimulatory and adhesion molecules such as CD86, CD70 and ICAM-1 was observed in HMCLs infected with LOAd700 and LOAd703, which encode for CD40L and CD40L/4-1BBL respectively. On the other hand, the apoptosis receptor Fas was upregulated post virus infection. The oncolytic effect of LOAd was then confirmed in a xenograft MM model injected intratumorally [57].

### 2.2. Non-Human Viruses

Some human viruses due to their pathogenicity and vaccination, which would neutralize their action, cannot be considered suitable as drugs. In recent years, the attention of researchers has focused on some non-human viruses that lack pathogenicity in humans, but are still capable of destroying human tumor tissue. These viruses exploit the same entry site and mechanism of human viruses to infect and kill tumor cells. Here we reported the non-human viruses considered safe platforms for the development of anti-MM oncolytic therapy. 

#### 2.2.1. Bovine Viral Diarrhea Virus

Recently, Marchica et al. demonstrated the efficiency of a bovine *Pestivirus*, bovine viral diarrhea virus (BVDV), in killing MM cells [58]. BVDV is a single-stranded RNA virus considered one of the major viral pathogens of cattle, directly associated with mucosal disease [59]. BVDV binds CD46 receptor, as reported for human MV, to infect MM cells [60]. The authors showed that BVDV treatment induces apoptosis selectively in myeloma cells through the activation of Caspase-3 in vitro. Interestingly, the oncolytic effect of BVDV was increased by Bortezomib pretreatment of HMCLs. Moreover, they confirmed the oncolytic BVDV activity in an in vivo subcutaneous myeloma mouse model, which showed a reduction of tumoral burden [58].

#### 2.2.2. Vaccinia Virus

Vaccinia virus (VV) is a linear double-stranded DNA virus, derived from the original cowpox or horsepox virus with an excellent history of safety in humans. Evidence of its oncolytic activity have been reported in in vitro studies using a VV depleted from thymidine kinase (TK) and vaccinia growth factor (VGF) to allow tumor specificity [61]. Deng et al. demonstrated that the VV cytopathic activity was specific for MM cells while no viral replication was observed in normal peripheral blood cells. The authors described a decrease of tumor volume and an increased survival of subcutaneous xenograft-bearing mice treated with VV [62]. More recently a study using two TK-deleted VV overexpressing respectively the antitumor factors, miR-34a and Smac, showed a synergistic induction of apoptosis through the activation of the caspase pathway [63]. Finally, Lei et al. have constructed a new VV overexpressing Beclin-1 gene (*VV-BECN1*), which induces in vitro autophagic myeloma cell death, but not apoptosis, through activation of sirtuin1 (*SIRT1*) [64].

#### 2.2.3. Myxoma Virus

Myxoma Virus (MYXV) is a non-human virus, with a non-segmented double-stranded DNA genome, which showed a significant oncolytic potential against several tumors, including MM [65]. Specifically, it has been demonstrated that MYXV discriminates myeloma cells from normal cells and subsequently eliminates these cells by inducing a rapid apoptotic response mediated by caspase-8 [66]. Moreover, a study from Dunlap et al. showed that MYXV infection causes the inhibition of activating transcription factor (*ATF*)4, primary mediator of apoptosis during the unfolded protein response (UPR), in MM cells [67].

## 3. Indirect Mechanisms of Action of Oncolytic Viruses in MM

The therapeutic efficacy of oncolytic viruses can also depend on indirect activation of the immune system against the tumor cells, through the infection of microenvironmental cells, which become carriers to deliver oncolytic viruses in the tumor site [68,69]. Once infection by oncolytic viruses has occurred, cancer cells can activate an antiviral response by releasing soluble factors that stimulate immune cells [70]. Activation of immune responses induced by oncolytic viruses creates an immunogenic microenvironment that increases cell death and tumor destruction [71]. MM microenvironment is characterized by T cell imbalances, functionally defective DCs, increased MDSC percentage and upregulation of immunosuppressive checkpoint proteins [2,72,73]. These features allow MM to evade host immune surveillance. In this context, oncolytic virotherapy is currently used as an immunotherapeutic approach to increase tumor immunogenicity.

Figure 2 summarizes the main indirect mechanisms.

### 3.1. Human Viruses

#### 3.1.1. Measles

A study conducted by Ong et al. used activated T cells as carriers for the oncolytic virus MV [74]. In particular, T cells, isolated from peripheral blood mononuclear cells (PBMCs), were loaded with previously modified MV to be directed towards the tumor sites and to escape neutralizing antibodies. Indeed, neutralizing antiviral antibodies can inhibit the anticancer activity of oncolytic viruses and in this case can prevent virus delivery to the target site. In this study, it was demonstrated that systemically administered T cells can act as carriers to deliver MV to MM subcutaneous plasmacytomas in severe combined immune deficient (SCID) mice. However, the treatment of immunized mice with MV immune serum inhibited the spread of the oncolytic MV to the tumor site. Nevertheless, oncolytic MV was detectable after the administration of serially diluted MV immune serum [74]. The authors suggest that good results can be obtained combining oncolytic therapy with plasmapheresis techniques or techniques that reduce pre-existing antiviral antibodies. Interestingly, the efficacy of MV vaccination is known to be inefficient in MM patients [75], and this could favor the use of oncolytic MV in these patients.

Other microenvironment cell types have shown promising results as virus delivery vehicles, able to access MM growth sites, such as mesenchymal progenitor cells and monocytes [76]. Peng et al. identified the presence of infiltrating CD68+, CD163+, S100- macrophages, and dispersed CD3+ T lymphocytes in plasmacytomas histologic sections from MM patients [77]. CD68+ macrophages, known as tumor-associated macrophages (TAMs) had a dendritic morphology and were in tight contact with several plasma cells. Since the TAMs are cells derived from monocyte lineage, isolated primary CD14+ at different stages of maturation were infected with attenuated Edmonston strain MV expressing enhanced GFP (MV-eGFP), or RFP (MV-RFP) or firefly luciferase (MV-Luc) in order to assess the ability to deliver the virus to MM cells. Mature DCs (mDCs) resulted as the most efficient cells among the different monocyte differentiation stages, while immature DCs (iDCs) showed moderate levels of infection. Subsequently, studying the biodistribution of these cells, the authors observed that the iDCs represented the best candidate to use as a vector of infection, in terms of cell viability. iDCs armed with MV were then inoculated intravenously in SCID mice, their distribution was evaluated by imaging and after 48 h of infection their presence was detected inside the tumor. These in vivo experiments showed that the survival of treated mice with iDCs as the MV carrier was significantly prolonged compared to the control group.

Recently, MV has been used as an immunotherapeutic agent to generate a persistent antitumor immune stimulation in MM patients [21]. In particular, a study by Packiriswamy et al. investigated primary T cell responses against a known panel of tumor-associated antigens (TAAs) in MM, after administration of oncolytic MV encoding sodium iodide symporter (MV-NIS) [78]. The T cell responses against the analyzed TAAs, specifically melanoma antigen gene (MAGE)-C1 and MAGE-A3, were increased after the MV-NIS treatment. Only one patient achieved complete remission with an advanced anticancer T-cell response. Another patient had mild focal relapses few months after oncolytic MV-NIS treatment, which were treated and resolved. The authors suggest that virotherapy could have potentiated the T-cell antitumor response, which was further improved after local radiotherapy treatment. Interestingly, MM patients treated with MV-NIS showed a significant increase in post-virotherapy CD3+ T lymphocytes, and this was largely due to an increase in CD8+ T cells; no changes were observed either in CD4+ T cells or in Tregs. Additionally, a more detailed analysis of the lymphocyte population showed an increased percentage of effector memory and central memory CD8+ after MV-NIS treatment, suggesting that virotherapy induced the activation of T lymphocytes.

#### 3.1.2. Adenovirus

Besides the direct effects reported in the above paragraph, Wenthe et al. described AdV additional activity on the tumor immune-microenvironment [57]. Specifically, the authors reported the upregulation of the costimulatory molecules, CD86 and human leukocyte antigen (HLA)-DR isotype, on the surface of MM cells upon LOAd infection. Moreover, coculture experiments with MM cells and healthy donor PBMCs showed an expansion of central memory T cells, along with the upregulation of the activation markers CD69, PD-1 and degranulation marker CD107a, especially after infection with LOAd703. Increased levels of IFN-γ were also detected suggesting the activation of T cell compartment. On the other hand, the authors showed that the percentage of Tregs was expanded by the presence of MM cells but it decreased after LOAd infection. Together these data suggest the ability of LOAd infection to increase T cell immune activation and thus revert the immune-suppressive effect of MM cells on the immune-microenvironment [57].

### 3.2. Non-Human Viruses

#### 3.2.1. Vesicular Stomatitis Virus

Among the enveloped single-stranded negative-sense RNA viruses with oncolytic activity, the vesicular stomatitis virus (VSV) is reported [79]. The distinguishing features of VSV are a small and easy to manipulate genome, its pantropism and the lack of pre-existing human immunity against VSV [80]. Surface molecules, such as low-density lipoprotein receptor (LDLR), are used by VSV for cell attachment [81]. LDLRs are ubiquitously expressed and this allows VSV to enter in different cell types [82]. However, its infection is normally inhibited by activation of protein kinase R (PKR) and IFN production [83,84]. Since the PKR system is defective in tumor cells, VSV has been shown to have a high selective lytic activity in these cells. In MM, preclinical studies have shown promising results in the use of VSV in vitro and in vivo. All reported data use genetically modified VSV to enhance its lytic effect, to monitor its infection, to improve host antitumor immunity and to maintain normal cells unaltered. In particular, a VSV able to express NIS was generated, enabling imaging and treatment with radioactive iodine. MM immunocompetent mice treated with VSV-NIS showed a reduction in the tumor burden. In addition, a further reduction was observed following treatment with radiotherapy [85]. Subsequently, Naik et al. inserted IFN-β into the VSV viral genome in order to improve the specificity of the infection and exploit the tumor immune signaling defects. Here the authors demonstrated that VSV-IFNβ has potent therapeutic efficacy against MM in immunocompetent mice, selectively and rapidly destroying tumors. Furthermore, intravenous administration of VSV-IFNβ in immunocompetent mice showed that a sufficient number of VSV-IFNβ virions reached the tumor site, carrying out its antitumor activity, and killed disseminated myeloma cells, increasing the survival of the mice. The authors suggest that IFNβ expression may promote late infiltration of immune cells into MM tumors, enhancing the antitumor immune response [86]. Subsequently, the same group determined that the optimal configuration for VSV is given by the insertion of IFNβ and NIS in its genome, to better perform its oncolytic activity. Treatment with VSV-IFNβ-NIS improved tumor reduction and survival of the mice compared with control VSV treatment. Furthermore, the immune-mediated eradication of minimal residual disease was observed in an immunocompetent MM mouse model. In fact, mice treated with VSV-IFNβ-NIS showed lower relapse rates than mice treated with VSV-IFNβ-NIS but depleted T lymphocytes [87]. Based on these results, a phase I trial was created to investigate the best dose and side effects of VSV-(h)IFNβ-NIS in the treatment of relapsed or unresponsive patients with MM, acute myeloid leukemia, or T-cell lymphoma (NCT03017820). The trial is still in the recruitment phase.

#### 3.2.2. Myxoma Virus

An indirect effect of MYXV in MM cells has been reported in a study from Villa et al., which demonstrated that T cells exposed to MYXV are better armed to kill MM cells, thus increasing the graft-versus-tumor. The mechanism behind this effect was the transfer of active oncolytic virus to residual cancer cells. On the other hand, MYXV infects activated T cells attenuating their proliferation and production of proinflammatory cytokines thus reducing the graft-versus-host disease. Based on these results, the authors suggest that the ex vivo virotherapy with MYXV may be a promising clinical adjunct to allo-hematopoietic cell transplantation regimens [88]. Similarly, Lilly et al. demonstrated that neutrophils from BM allo-transplanted mice infected with MXYV potentially act as carrier cells to target and eradicate residual myeloma cells.

## 4. Oncolytic Viruses in Combination with Anti-MM Drugs

Several studies reported that oncolytic viruses can be used to enhance the action of other drugs [89]. Among these, RV is used in combination with drugs currently in use in MM clinical practice. IMiDs are the backbone regimens in the treatment of MM; among these, lenalidomide enhances T-cell and NK-cell activation and reduces regulatory T-cell function [90]. A study conducted by Parrish et al. demonstrated that the combination of RV and lenalidomide and/or dexamethasone induces both cell killing direct effect and an antitumor immune response effect on MM cells. In particular, they observed that the cell death increases in CD138+ cells from MM patients treated with lenalidomide and/or dexamethasone adding RV. Moreover, the treatment of PBMCs from MM patients with RV upregulated CD69 expression on NK cells, CD4+ T cells, CD8+ T and NK showed enhanced degranulation against autologous tumor cells [91]. 

Another emerging class of drugs for MM treatment are histone deacetylase inhibitors (HDACi) that alter the regulation of histone and non-histone proteins, inducing the expression of proapoptotic proteins [92]. A study from Stiff et al. showed that RV treatment in combination with HDACi enhances the antitumor therapeutic effect of the oncolytic virus. It was observed that JAM-1, the RV receptor, was increased after treatment of MM cells with clinically used doses of HDACi. Indeed, it was reported that HDACi increase the expression of JAM-1 by epigenetically regulating its promoter. For this reason, in vitro and in vivo experiments show that the combined treatment of HDACi and RV synergistically increases the killing of MM cells. Although Entinostat and AR-42 show a greater synergic activity with RV, overall the HDACi tested showed a synergistic effect with oncolytic RV [93].

Some combination strategies are focused on potentiating the anticancer effects of immune-check point inhibitors, such as anti-PD-1 and anti-PD-L1. Kelly et al. reported that the RV treatment of MM cells increases the expression of PD-L1 [94]. This upregulation could be used as a precision strategy to potentiate the anti-MM efficacy of immune-checkpoint inhibitors, which have not shown clinically meaningful results when used in monotherapy [95] in part because of a heterogeneous PD-1/PD-L1 expression among patients with MM [96,97]. In a study using a xenograft mouse model, RV treatment produced significant antitumor activity against MM cells, promoting the viral replication in tumors. Moreover, the treatment with RV induced an upregulation of PD-L1 expression in MM cells. In line with these observations, the major antitumoral effect was obtained pretreating mice with RV, followed by anti-PD-L1 administration. Indeed, RV must infect and replicate inside MM cells in order to increase PD-L1 expression. Mice treated with the combination of the RV and anti-PD-L1 antibody displayed a disease regression with a decrease in M protein levels and an increase of mice survival. These findings thus provide the basis to test the combination of RV and immune-checkpoint inhibitors treatment in MM patients in future clinical trials.

Another therapeutic combination studied in the context of MM was the treatment of RV and bortezomib, the first-generation PI that induces cell-cycle arrest and apoptosis in myeloma cells [44,98]. In this study, the anti-MM effect resulted in an accumulation of viral and ubiquitinated proteins, which led to increased ER stress, NOXA induction and apoptosis. Kelly et al. showed that in a xenograft and syngeneic bone disease mouse model of MM, the combination of RV infection and proteasomal inhibition by Bortezomib reduced the tumor burden. Another study from Thirukkumaran et al. showed that the combination treatment with RV and bortezomib leads to significant immune modulation, increasing CD8+ and CD4+ tumor infiltrating lymphocytes (TILs), activated NK-T cells while reducing Tregs and TAMs, thus reversing myeloma-induced immune suppression [99]. An expansion of memory T cells, which are pivotal players in eradicating relapsed disease, was further observed. Moreover, a significantly enhanced expression of IFN-β and IFN-γ was described, reflecting an enhanced immune activity. The authors showed that RV monotherapy and RV in combination with bortezomib significantly upregulates the expression of immune-check point molecules, such as PD-L1, PD-L2 and indoleamine 2,3-dioxygenase (IDO)-1 in myeloma cells from both the BM and spleen in syngeneic murine model [99]. In light of these data, a phase Ib study of RV in combination with bortezomib and dexamethasone was conducted in patients with relapsed/refractory MM (NCT02514382) [100]. In a previous study, RV was well tolerated in patients with relapsed refractory MM and was associated with prolonged stable disease [101]. The patients well tolerated the RV treatment in combination with bortezomib and dexamethasone, without severe toxicities. However, only one patient remained on treatment and pending results will complete the study.

These combination treatments could potentially play an important role in enhancing the clinical therapeutic efficacy of systemic RV in patients with MM.

New perspectives in the cure of the disease including CAR-T cells, drug-conjugate mAbs and bispecific Abs have been recently developed for MM treatment. However, the use of these agents in combination with oncolytic viruses has been explored only in solid tumors [102,103]. In this setting there are some evidences that, for example, the combination of oncolytic virus and CAR-T could improve patients outcome boosting immune system and facilitate the CAR-T trafficking [104,105]. These interesting results leave an open question on the efficacy of these combination strategies in MM.

The main features of oncolytic viruses cited in this review are reported in Table 1.

## 5. Conclusions

To conclude, oncolytic viruses represent a pharmacologic approach capable of achieving tumor killing through the combination of direct and indirect mechanisms in order to inhibit cancer progression. Direct destruction is a consequence of the selective replication of the virus in cancer cells, which damages them by triggering cell death. Indirect tumor destruction, on the other hand, results from the ability to modulate the interaction between tumor and microenvironment cells. In particular, viral infection induces inflammatory processes that activate adaptive antiviral immune responses targeting the immune-microenvironment cells and addressing the cytotoxic responses of T lymphocytes. Moreover, the use of oncolytic viruses activates T cells against the antigens present on the surface of non-infected tumor cells thus eliminating residual disease. These mechanisms thus highlight the ability of virotherapy to overcome the immune suppression, which is the hallmark of MM patients, and to reactivate their immune system against tumor cells. To date, the results from studies using oncolytic therapy for MM treatment are encouraging without severe side effects.

Several strategies should be considered in the choice of an appropriate virus against a specific tumor target in order to improve the oncolytic effect and limit adverse effects. Targeting can be achieved based on natural tropisms to specific tissues or cell types or by engineering these tropisms. For example some oncolytic viruses use cell surface molecules for entry that are abnormally upregulated in cancer cells, such as CD46 for MV, which is frequently overexpressed to avoid immune system recognition and elimination. Similarly, overexpression of ICAM-1 and DAF by MM cells is used as entry for CV. Another strategy to enhance the oncolytic efficacy is “arming” viruses with suicide genes to increase their ability to directly kill cancer cells.

Additional factors have to be considered for oncolytic viral treatments. Indeed, the vaccination or natural infections of patients could represent a limitation for the use of human viruses as possible neutralizing antibodies may already exist against certain viruses, and further antibodies could be produced more rapidly because of immunological memory. A possible mechanism to avoid this neutralization includes the encapsulation of oncolytic viruses in polymer coatings to ensure viral replication and circulation [106]. Moreover, another approach based on alternative non-human viruses has been developed to bypass this issue.

Finally, oncolytic viruses could represent, as monotherapy or in combination with other drugs, a promising strategy for the treatment of patients who do not benefit from the currently available drug therapies in myeloma.

## Figures and Tables

**Figure 1 ijms-22-02259-f001:**
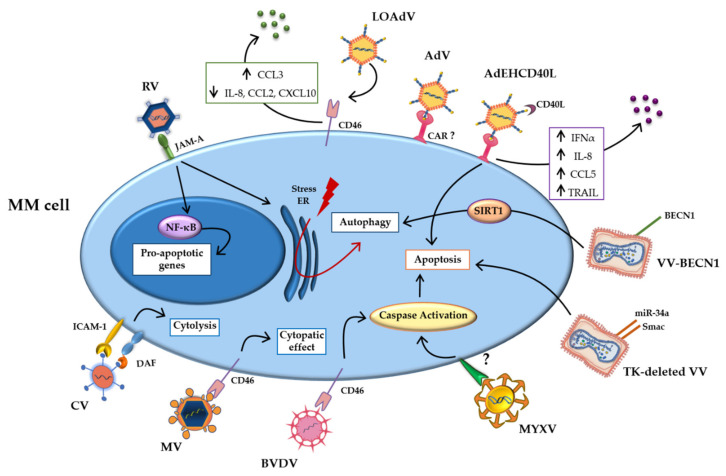
Direct mechanisms of oncolytic viruses. Schematic representation of direct mechanisms of the oncolytic viruses studied in multiple myeloma (MM) cells, showing the main pathways activated by oncolytic infection. Abbreviations: AdV: Adenovirus; BECN-1: Beclin-1; BVDV: Bovine viral diarrhea virus; CAR: coxsackievirus and adenovirus receptor; CCL2: Chemokine (C-C motif) ligand 2; CCL3: Chemokine (C-C motif) ligand 3; CCL5 Chemokine (C-C motif) ligand 5; CD40L: CD40-ligand; CXCL10: C-X-C motif chemokine ligand 10; CV: Coxsakie virus; DAF: decay-accelerating factor; ER: Endoplasmic reticulum; ICAM-1: intercellular adhesion molecule; INFα: Interferonα; IL-8: Interleukin-8; JAM-A: Junctional adhesion molecule-A; LOAd: Lokon oncolytic Adenovirus; MM: Multiple myeloma; MYXV: Myxoma Virus; MV: Measles virus; NF-κB: Nuclear factor kappa-light-chain-enhancer of activated B cells; NIS: Sodium iodide symporter; RV: Reovirus; SIRT1: Sirtuin1; TK: thymidine kinase; TRAIL: TNF-related apoptosis-inducing ligand; VV: Vaccinia Virus.

**Figure 2 ijms-22-02259-f002:**
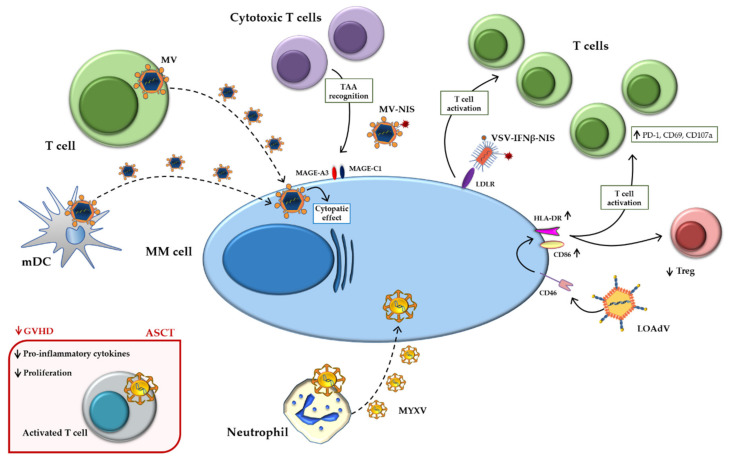
Indirect mechanisms of oncolytic viruses. Schematic representation of indirect mechanisms of oncolytic viruses using immune-microenvironment cells as carriers and active players against MM. Abbreviations: ASCT: Autologous stem cell transplantation; HLA-DR: Human leukocyte Antigen-DR isotype; GVHD: Graft versus host disease; INFβ: Interferonβ; LDLR: Low-density lipoprotein receptor; LOAd: Lokon oncolytic Adenovirus; MAGE: Melanoma antigen gene; mDCs: Mature dendritic cells; MM: Multiple myeloma; MYXV: Myxoma Virus; MV: Measles virus; NIS: Sodium iodide symporter; PD-1: Programmed death-1; TAA: Tumor-associated antigen; Treg: Regulatory T cell; VSV: Vesicular stomatitis virus.

**Table 1 ijms-22-02259-t001:** Human and non-human oncolytic viruses currently studied for multiple myeloma therapy.

	Oncolytic Virus	Genome	Direct Mechanism	Indirect Mechanism	Data
Human viruses	*Measles*	ss(−)RNA	Cytopathic effect	T and mDC cells used as carriers to induce cytopathic effect Increase of T cells responses against TAAs	**Pre-clinical:** anti-tumorigenic and anti-neoplastic activity in xenograft models. **Clinical:** decrease in serum FLC levels, reduction of the percentage of malignant plasma cells and of extramedullary masses in refractory MM patients. Increase of CD8+ cells after MV treatment.
	*Reovirus*	dsRNA	Activation of NF-κB Apoptosis Autophagy ER stress	-	**Pre-clinical:** anti-tumor activity in a xenograft mouse model. **Clinical:** associated with prolonged stable disease in relapsed refractory MM patients. **Combination therapy:** lenalidomide and/or dexamethasone; HDACi; anti-PD-L1; bortezomib and/or dexamethasone.
	*Coxsackie virus*	ss(+)RNA	Cytolysis	-	**Pre-clinical:** killing of pre-malignant plasma cells.
	*Adenovirus*	dsDNA	Inhibition cell growth Cytokines release Apoptosis	Up-regulation of co-stimulatory molecules Expansion of central memory T cells Decrease of T regs	**Pre-clinical:** reduction of tumor volume in a xenograft model.
Non-human viruses	*Bovine viral diarrhea virus*	ssRNA	Apoptosis	-	**Pre-clinical:** reduction of tumor burden in mouse model.
	*Vaccinia virus*	dsDNA	Cytopathic effect Apoptosis Autophagy	-	**Pre-clinical:** decrease of tumor volume and an increased survival of subcutaneous xenograft-bearing mice.
	*Myxoma virus*	dsDNA	Apoptosis	T cells and neutrophils used as carriers to eradicate residual MM cells	**Pre-clinical:** reduction of graft versus host disease, attenuation of T cell proliferation and production of pro-inflammatory cytokines.
	*Vesicular stomatitis virus*	ss(−)RNA	-	T cells activation	**Pre-clinical:** reduction of tumor burden and immune-mediated eradication of minimal residual disease in MM immunocompetent mice.

**Abbreviations:** ds: double stranded; ER: endoplasmic reticulum; FLC: free light chain; HDACi: histone deacetylase inhibitors; mDC: mature dendritic cell; MM: multiple myeloma; MV: measles virus; NF-κB: Nuclear factor kappa-light-chain-enhancer of activated B cells; ss: single stranded; TAA: Tumor-associated antigen; T regs: Regulatory T cells.

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
