# Peer review of "Oncolytic Virotherapy and Microenvironment in Multiple Myeloma"

_ijms, 2021, doi:10.3390/ijms22052259_

Round 1

Reviewer 1 Report

The manuscript by Marchica and co-authors describes an effort to give a review of available oncolytic virotherapy for multiple myeloma patients.

The authors did a great job to distinguish human vs non-human viruses and their direct mechanism and mechanism on the microenvironment.  Besides minor English/technical errors to polish, I would suggest to authors to change the paragraph where they are talking about their own findings, and instead of 'our group findings' give a reference to the manuscript such as Marchica et al. (). My other comment is regarding 'oncolytic viruses represents a new approach' in the conclusion. The authors are encouraged to change this statement, as this is not a novel approach. The approach is dating from the last century, with approvals for use going back 15 yrs ago. If the authors meant specifically for MM patients, then they should say so.

Overall, very well written, in a concise manner. 

Author Response

Reviewer1 :

The manuscript by Marchica and co-authors describes an effort to give a review of available oncolytic virotherapy for multiple myeloma patients. The authors did a great job to distinguish human vs non-human viruses and their direct mechanism and mechanism on the microenvironment.  Besides minor English/technical errors to polish, I would suggest to authors to change the paragraph where they are talking about their own findings, and instead of 'our group findings' give a reference to the manuscript such as Marchica et al. (). My other comment is regarding 'oncolytic viruses represents a new approach' in the conclusion. The authors are encouraged to change this statement, as this is not a novel approach. The approach is dating from the last century, with approvals for use going back 15 yrs ago. If the authors meant specifically for MM patients, then they should say so. Overall, very well written, in a concise manner. 

We thank the Reviewer for the kind reply. We have modified the manuscript as suggested and edited grammar mistakes.

Reviewer 2 Report

Multiple myeloma treatment is challenging. The development of therapeutic regimens over the past decade that incorporates the proteasome inhibitor bortezomib and the immunomodulatory drugs thalidomide and lenalidomide has been the cornerstone of improving the outcome of patients with myeloma. Although these treatment regimens have improved patient survival, nearly all patients eventually relapse. Therefore, new and more effective treatment options and their combinations are highly needed. 

The review by Marchica et al. describes the use of oncolytic vectors in myeloma therapy. Authors also discussed combinatory therapies with other myeloma targeting drugs.

The following points require further attention:

  • Figure 1: Note that not all mentioned abbreviations in the Figure have been explained in the caption section.
  • Both Figure 1 and 2 should contain an explanation, what they are presenting in general. Having a short title with a list of abbreviations is not an easy way to follow the graphic. Maybe a table presenting various molecular pathways, the vector list can help to visualize and list down the features of the direct and indirect mechanism of OVs.
  • I suggest adding a Table where stating mentioned in the text various oncolytic vectors, their basic characteristics,  major findings in both pre-clinical and clinical studies. 
  • I suggest also to add information on the standard of care therapies in myeloma, its efficacy, limitations. 
  • Oncolytic vectors are mainly used in solid cancer therapies. Due to their immunogenicity, presence or development of neutralizing antibodies... are not primarily administered intravenously (but rather intratumorally). Therefore, please elaborate more info on genetic modifications, strain selection, making oncolytic vector more efficacious. 
  • Please provide more detailed elaboration on current limitations in regards to the use of oncolytic vectors in MM therapy, what is the current challenge in the field?
  • Would be also important to add a paragraph where focusing on a combinatory therapy of OV with  CAR-T cells, drug-conjugate mAbs and bi- 41 specific Abs.

Author Response

Reviewer2:

Multiple myeloma treatment is challenging. The development of therapeutic regimens over the past decade that incorporates the proteasome inhibitor bortezomib and the immunomodulatory drugs thalidomide and lenalidomide has been the cornerstone of improving the outcome of patients with myeloma. Although these treatment regimens have improved patient survival, nearly all patients eventually relapse. Therefore, new and more effective treatment options and their combinations are highly needed. 

The review by Marchica et al. describes the use of oncolytic vectors in myeloma therapy. Authors also discussed combinatory therapies with other myeloma targeting drugs.The following points require further attention:

  • Figure 1: Note that not all mentioned abbreviations in the Figure have been explained in the caption section.

All the abbreviations are now explained in the revised manuscript.

  • Both Figure 1 and 2 should contain an explanation, what they are presenting in general. Having a short title with a list of abbreviations is not an easy way to follow the graphic. Maybe a table presenting various molecular pathways, the vector list can help to visualize and list down the features of the direct and indirect mechanism of OVs.

A short description has been included for each figure in the revised manuscript.

  • I suggest adding a Table where stating mentioned in the text various oncolytic vectors, their basic characteristics, major findings in both pre-clinical and clinical studies.

We thank the Reviewer for the suggestion. A comprehensive table reporting the main characteristics of both human and non-human viruses has been added. 

  • I suggest also to add information on the standard of care therapies in myeloma, its efficacy, limitations. 

As suggested by the Reviewer, we added a brief information of standard of care in the treatment of multiple myeloma in the introduction section, since a detailed and comprehensive information of the therapy of multiple myeloma patients is out of the scope of the article

  • Oncolytic vectors are mainly used in solid cancer therapies. Due to their immunogenicity, presence or development of neutralizing antibodies... are not primarily administered intravenously (but rather intratumorally). Therefore, please elaborate more info on genetic modifications, strain selection, making oncolytic vector more efficacious. Please provide more detailed elaboration on current limitations in regards to the use of oncolytic vectors in MM therapy, what is the current challenge in the field?

As requested by Reviewer we provided more details about the limitation of using oncolytic viruses and different strategies to enhance their activity in the conclusions section. 

  • Would be also important to add a paragraph where focusing on a combinatory therapy of OV with  CAR-T cells, drug-conjugate mAbs and bi- 41 specific Abs

Despite the interesting results of OV combination with CAR-T cells, drug-conjugate mAbs and bi-specific Abs in solid tumors, no literature data are currently available in MM, to our knowledge. However, we have reported some details about these combinations in solid tumors which highlight the need to explore this interesting question also in multiple myeloma.

Round 2

Reviewer 2 Report

The authors provided satisfactory replies and corrections.